

# Greenhouse gas emissions from riparian zone cropland in a tributary bay of the Three Gorges Reservoir, China

XiaoXiao Wang[1,2], Ping Huang[1], Maohua Ma[1], Kun Shan[1], Zhaofei Wen[1] and Shengjun Wu[1]

[1] Key Laboratory of Reservoir Aquatic Environment, Chongqing Institute of Green and Intelligent Technology, Chinese Academy of Sciences, ChongQing, China
[2] University of Chinese Academy of Sciences, Beijing, China

## ABSTRACT

**Background**. A huge reservoir was formed by the Three Gorges Dam in China, which also formed a riparian zone along the bank of the reservoir. In the period of low water-level, the riparian zone in tributary bays of the Three Gorges Reservoir (TGR) was always unordered cultivated, owing to its gentle slope and high soil fertility. This land-use practice creates high potential of generating greenhouse gas (GHG) emissions with periodic water level fluctuation.

**Methods**. To evaluate potential GHG emissions from the soil-air interface, the static opaque chamber method was adopted to evaluate the effect of elevations (180 m, 175 m, 170 m and 165 m) and land use types (dry lands, paddy fields and grass fields) from April to September in 2015 and 2016.

**Results**. The results showed that carbon dioxide ($CO_2$) was the main contributor of GHG emission in riparian zone most likely because of high organic carbon from residues. Furthermore, high soil water content in paddy fields resulted in significantly higher methane ($CH_4$) flux than that in dry lands and grass fields. Compared to grass fields, anthropogenic activities in croplands were attributed with a decrease of soil total carbon and GHG emissions. However, inundation duration of different elevations was found to have no significant effect on $CH_4$ and $CO_2$ emissions in the riparian zone, and the mean nitrous oxide ($N_2O$) flux from dry lands at an elevation of 165 m was significantly higher than that of other elevations likely because of tillage and manure application. The high $N_2O$ fluxes produced from tillage and fertilizer suggested that, in order to potentially mitigate GHG emissions from the riparian zone, more attention must be paid to the farming practices in dry lands at low elevations (below 165 m) in the riparian zone. Understanding factors that contribute to GHG emissions will help guide ecological restoration of riparian zones in the TGR.

## INTRODUCTION

The report from the Intergovernmental Panel on Climate Change (IPCC) pointed out that carbon dioxide ($CO_2$), methane ($CH_4$) and nitrous oxide ($N_2O$) are the three most

Corresponding authors
XiaoXiao Wang,
wangxiaoxiao@cigit.ac.cn
Shengjun Wu, wsj@cigit.ac.cn

important greenhouse gases (GHGs) and can remain ten to hundreds of years (or even more) in the atmosphere. Although the research on GHG emissions from reservoirs started in the 1990s (*Kelly & Hecky, 1993*), extra-large reservoir ecosystems have been less studied in China. Around the world, such studies have mostly focused on a permanent flooded area, rather than the riparian zone (*Chen et al., 2014*; *Duchemin et al., 1995*; *Huttunen et al., 2003*; *Kelly et al., 1994*).

As the ecotone area of substance and energy exchange between aquatic and terrestrial ecosystems, the riparian zone is regarded as a hot region of GHG emissions, although the area is relatively smaller than reservoirs (*Gilliam, 1994*; *Kankaala, Ojala & Käki, 2004*; *Naimanand & Décamps, 1997*). At present, theories for marsh wetland and paddy fields, as well as lakes and rivers, are usually applied in order to understand the GHG emissions and their mechanisms of riparian zones (*Demarty & Bastien, 2011*). Compared with natural wetlands and paddy fields, the riparian zones of reservoirs are characterized with high and unorderly fluxes owing to frequent water level fluctuations. First, dry-wet alternation events can disintegrate the soil aggregates, changing the soil porosity to release the nutrients (*Park, Sul & Smucker, 2007*). Second, the decomposed plant residues, which resulted in soil organic matter accumulation after submergence (*Lima et al., 2008*), may increase $CO_2$ and $CH_4$ emission. Third, cultivation as a typical anthropogenic activity in the riparian zone not only changes soil structure, but also alters processes of nutrients cycling (*Kasimirklemedtsson et al., 2010*). Agricultural GHG emissions have been estimated as 11% of China's national emissions i.e., 820Mt $CO_2$-equivilent, whereas emissions from rice cultivation were 374 Mt $CO_2$-equivilent (*National Coordination Committee on Climate Change, 2012*). Most of the research on GHG emissions in cropland focus on the field of permanent reclamation, while studies on intermittent reclamation in the riparian zones with periodic water level fluctuation are lacking.

The Three Gorges Dam, regarded as the biggest hydro-power project in the world, has been operating at full capacity since the end of 2010. Since then, a large area of land has been submerged into reservoir water. The water level of Three Gorges Reservoir (TGR) fluctuated from the elevation of 145 m to 175 m (Fig. 1). A total of 632 km$^2$ of land were submerged due to the dam construction, and approximately 33.1% of total inundated area was farm land in Chongqing (*Xu, 2013*). Bay ecosystem is the typical zone in TGR, and is directly influenced by anthropogenic activities and water level fluctuation, because of its low slope, large area and high density of population nearby (*Cheng et al., 2017*). Most of the GHG emission studies on the TGR in recent years focused on the air-water interface for eutrophication, ignoring the soil-air interface in the riparian zone (*Chen et al., 2011*; *Jiang et al., 2012*). The riparian zone as a newly created marsh was in the transition stage from terrestrial ecosystem to wetland ecosystem, and became an important source of $CH_4$, $CO_2$ and $N_2O$ emissions, owing to the intense biogeochemistry (*Fisher et al., 2014*; *Vidon et al., 2016*). Farming practice was found to play an important role in $CH_4$ and $N_2O$ emissions during various water levels in the TGR, after normal impoundment in 2010 (*Yang et al., 2012*; *Chen et al., 2011*). As the number of fluctuation periods increased, the quantity and

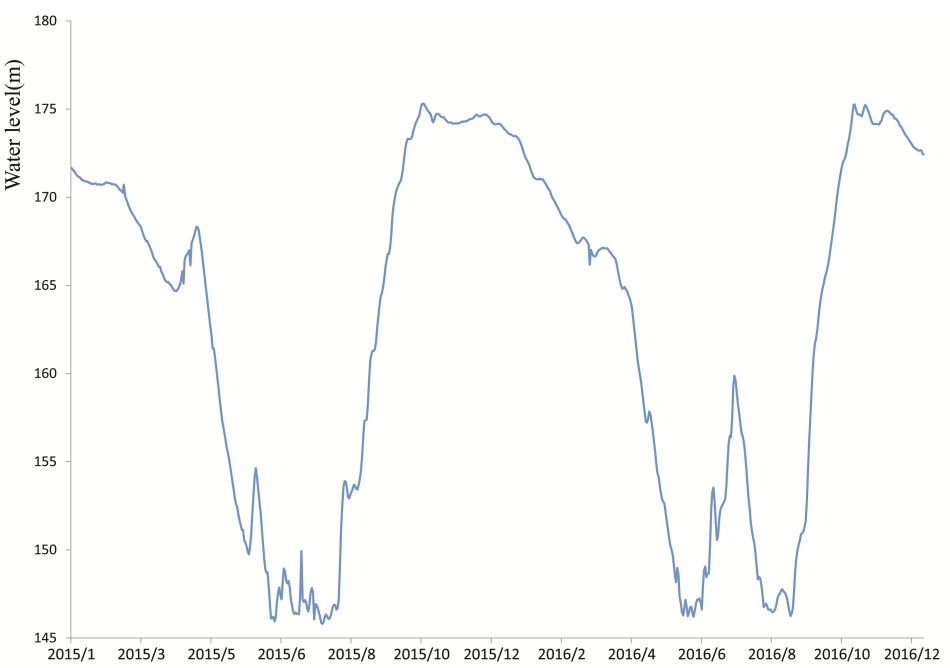

**Figure 1** **Water level fluctuation of Three Gorges Reservior from 2015 to 2016.**

quality of soil organic matter in agricultural ecosystems would change in the riparian zone (*Demarty & Bastien, 2011*), potentially leading to a change of the GHG emissions.

In summary, the study of $CO_2$, $CH_4$ and $N_2O$ emissions in cropland of the riparian zone in the TGR addresses a data gap of GHG emissions influenced by both anthropogenic activities and water level fluctuation. This study was motivated by two research questions: (1) Is cultivation, as the typical human activity in the bay riparian zone, a key factor in producing the GHG emission after several fluctuation periods? (2) How do cultivation and water level fluctuation jointly influence cropland emission in the riparian zone? To answer questions, three aims were set up as follows: (1) comparing flux differences of $CO_2$, $CH_4$ and $N_2O$ in different elevations in riparian cropland; (2) comparing flux differences of $CO_2$, $CH_4$ and $N_2O$ among dry lands, paddy fields and grasslands in the riparian zone during the growing season; (3) evaluating the Global Warming Potential of the cropland in the riparian zone.

## MATERIALS & METHODS

### General description of farm land and experimental design

The study area was located at the Wuyang Bay (31°11′20″N, 108°27′40″E) in the Yetang Creek of the Pengxi river basin in the TGR (Fig. 2A). The total area is ca.2.44 × 10⁵m², with a mean annual air temperature of 18.2 °C (*Chen et al., 2014*). According to hydrologic records, the water level of this area rises to 175m at the end of October (the reservoir charging at peak level) and drops to 155 m at early April as the reservoir discharges. Thus, agricultural activities in this area are carried out between April to August. In this study, an

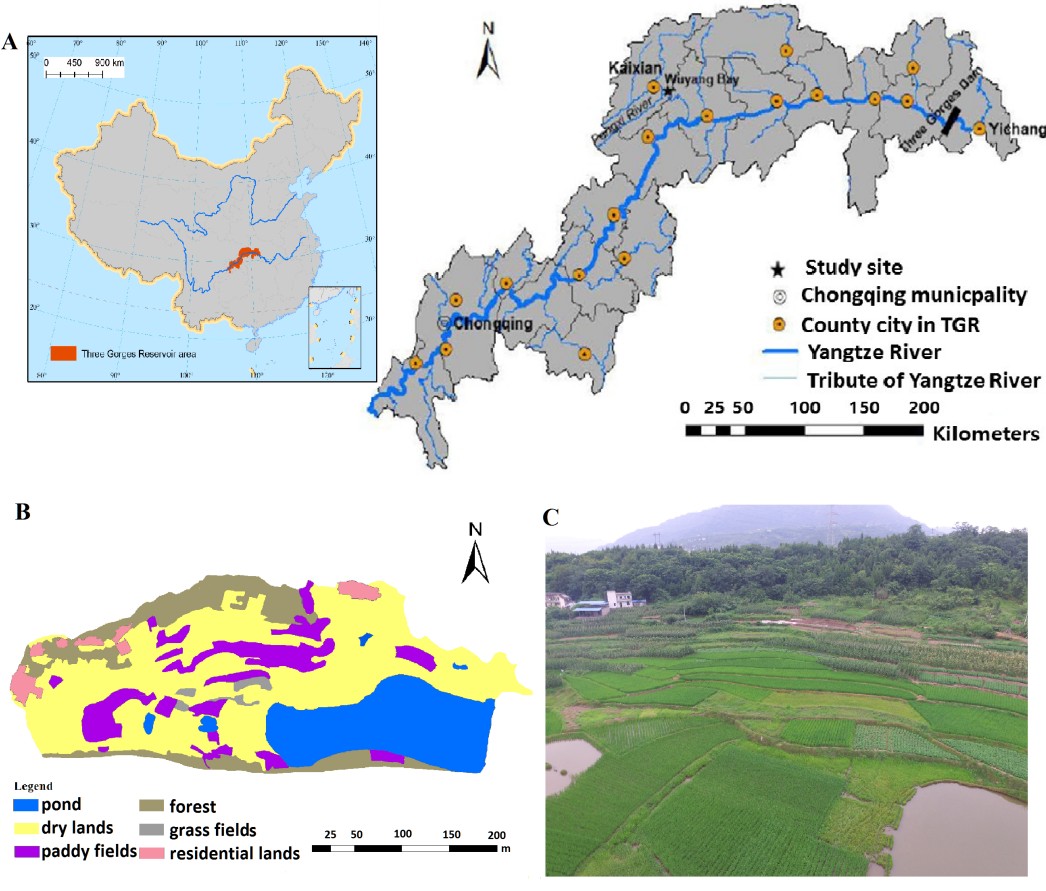

**Figure 2  The location and land use status of Wuyang Bay.**

area from 165 m to 180 m during the crop growing season was chosen to analyze GHG emissions.

According to field surveys in 2015, a total of 25 households with a population of 106 were recorded, and the main income source is agricultural production in Wuyang Bay. The soil type of this area is yellow soil (soil taxonomy), and the basic physic-chemical properties of experimental soil are shown in Table 1. Main crops in the study area are corn in dry lands and rice in paddy fields, and some vegetables at an elevation of 165 m (Figs. 2B and 2C). Some abandoned croplands were not used for several years, and have gradually become grass fields. Fertilizers were manually applied immediately after corn planting, and additional fertilizers were applied to paddy fields in March. Amounts of the fertilizers were about 1,500 kg hm$^{-2}$ a$^{-1}$ of urea and 1,500 kg hm$^{-2}$ a$^{-1}$ of compound fertilizer in corn fields, and about 1,050 to 1,500 kg hm$^{-2}$ a$^{-1}$ of urea and 1,050 kg hm$^{-2}$ a$^{-1}$ of compound fertilizer in paddy fields. Manure was applied in vegetable fields. After crop harvest, most rice straw was cut and moved to outside fields and corn straw was left in the field.

**Table 1  Physical and chemical properties of soil collected from riparian zone of Wuyang bay in 2015.**

| Land use | Soil organic matter (g kg$^{-1}$) | Bulk density (g cm$^{-3}$) | pH | Total nitrogen (mg kg$^{-1}$) | NH$_4^+$-N (mg kg$^{-1}$) | NO$_3^-$-N (mg kg$^{-1}$) |
|---|---|---|---|---|---|---|
| Paddy fields ($n = 6$) | $17.16 \pm 4.04$ | $1.12 \pm 0.14$ | $7.14 \pm 0.27$ | $1.13 \pm 0.35$ | $17.21 \pm 17.90$ | $11.45 \pm 24.06$ |
| Dry lands ($n = 6$) | $13.36 \pm 2.29$ | $1.34 \pm 0.12$ | $7.19 \pm 0.32$ | $1.56 \pm 0.31$ | $14.39 \pm 21.12$ | $17.21 \pm 17.90$ |
| Grass fields ($n = 6$) | $16.90 \pm 3.74$ | $1.38 \pm 0.08$ | $7.12 \pm 0.24$ | $1.52 \pm 0.29$ | $6.13 \pm 4.44$ | $3.43 \pm 3.23$ |

**Notes.**

Values are means ± standard error.

n, the sample size.

## Field sampling and measurements

Based on variations in the water levels and different land-use types, the experimental sites were chosen and classified into three groups: (1) different elevations (165–180 m) of croplands; (2) different land use types of dry lands, paddy fields and grass fields; (3) diurnal variations in dry lands and paddy fields.

Soil surface gas samples were collected almost monthly using the static-chamber method and a modified gas chromatograph method from April through September in 2015 and 2016. The closed static chamber ($\varphi$30.5 cm) (Fig. 3) was an opaque PVC material that was mounted on a PVC base frame in each plot, which had been inserted into the soil to a 10 cm depth. To make the covers gastight, water was poured into the grooves of the frame before each gas sampling. Each sampling site in every elevation was represented by one static chamber in dry land and one in paddy field in 2015. Three replicate static chambers were set up in dry lands, paddy fields and grass fields separately at the elevation of 170 m in 2016 for land use analyses. In each plot, GHG flux were made at the row position between plants in croplands, and aboveground vegetation in each plot of grasslands was cut off before gas sampling. Sampling time was selected from 9 a.m. to 11 a.m., when transient emissions represent the mean emission of the day (*Li et al., 2016*). Gas samples were collected from each chamber at 0, 10, 20 and 30 min after the chamber was deployed using a polypropylene medical syringe through a triple valve.

Samplings to assess diurnal variations of GHG emissions were conducted in May, July and September. May and July represent the growing season and September represents the harvest season. One sampling site in dry land and paddy field at 170 m was chosen to measure the diurnal GHG emissions. Gas samples were collected every 2 h from 8:00 of the first day to 6:00 of the subsequent day in Beijing standard time.

The concentration of N$_2$O, CH$_4$ and CO$_2$ in gas samples were determined by gas chromatography (Agilent 7890B). CO$_2$ andCH$_4$ were detected with a hydrogen flame ionization detector (FID). N$_2$O was detected with a Porapak Q packed column (80/100 mesh) and electron-capture detector (ECD) with a 63Ni. The temperature of the detector was 300 °C. Fluxes of N$_2$O, CH$_4$ and CO$_2$ were calculated as by the algorithm follows (*Flessa, Dörsch & Beese, 1995*):

$$F = \rho * \frac{V}{A} * \frac{P}{P_0} * \frac{273}{273 + T} * \frac{dc}{dt} \tag{1}$$
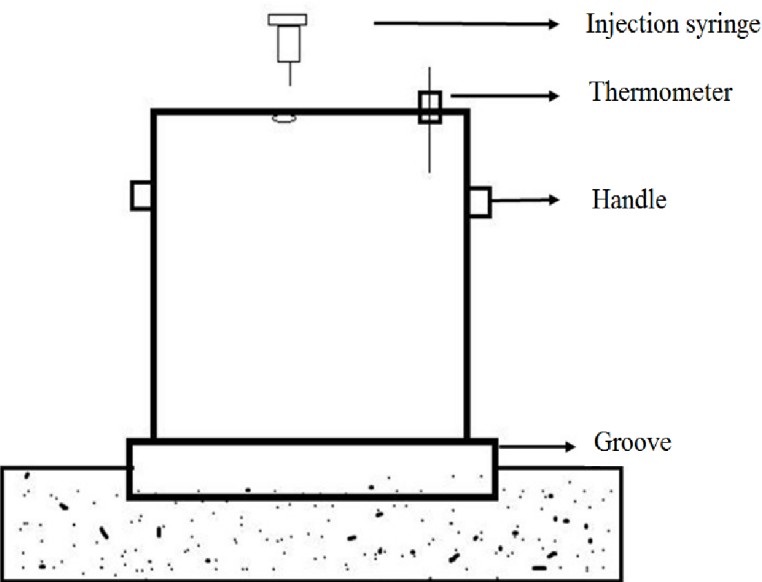

**Figure 3 Diagram of sampling chamber used in this study.**

Where $\rho$(kg m$^{-3}$) is the concentration of GHG in standard atmospheric condition; V(m$^3$) is the air volume in the chamber; $A$ (m$^2$) is the soil surface area in the chamber; $P$ (Pa) is atmospheric pressure; $T$ (°C) is the air temperature in the chamber; dc/dt is concentration changes of GHGs in chamber per unit time (ppm h$^{-1}$). $F$ is the flux of GHGs (mg m$^{-2}$ h$^{-1}$).

Total daily flux (TDF) was calculated using the following formula (*Xu et al., 2017*):

$$TDF = \left( \sum_{i=1}^{n=12} F_i \right) \times 2 \tag{2}$$

Where $i$ is the measuring time and $F_i$ means the flux at different times of the day.

Cumulative emission of GHG (CE) was calculated as follows:

$$CE = \sum_{i=1}^{n} \left( \frac{F_i + F_{i+1}}{2} \right) * (t_{i+1} - t_i) * 24/100 \tag{3}$$

Where $CE$ is the cumulative fluxes of GHG (kg hm$^{-1}$); $F$ is the flux of GHG; $i$ is the number of sampling events; $t$ is the day of the sampling events; $n$ is the total number of sampling events. Cumulative fluxes of GHG in paddy fields and dry lands were calculated from April to September in 2015 and 2016.

During each gas sampling event, soil volumetric water content (VWC) and soil temperature (ST) (10 cm depth) were measured within a radius of 30 cm from the sampling chamber using a portable moisture meter (HH2) and thermometer in soil. Soil gravimetric water content (GWC), nitrate nitrogen (NO$_3^-$-N), ammonia nitrogen (NH$_4^+$-N), total carbon (TC), total nitrogen (TN) and dissolved organic carbon (DOC) were measured for the topsoil (0–15 cm) within a radius of 30 cm from the sampling

chamber. Soil samples were taken to the lab and stored at 4 °C for the physicochemical analysis. $NO_3^-$-N and $NH_4^+$-N were measured by a flow injection analyzer (FIA star 5000); TC and TN were measured with an elemental analyzer (VarioEL cube). PH was measured after shaking with water for 30 min (*Ye et al., 2016*), soil organic matter was measured by LOI550 (*Zhou et al., 2014*), soil GWC was determined using the drying method (105 °C for 12 h), and bulk density was determined in the gravimetric method (*Avnimelech et al., 2001*). Dissolved organic carbon was measured through colorimetric analysis (*Yang et al., 2016*).

## Statistical analysis

The Global Warming Potential (GWP) was developed to reflect the global warming impacts of different gases. To reflect the total GHG emissions, the $CH_4$ and $N_2O$ fluxes were converted to a $CO_2$-equivalent with unit $mg\text{-}CO_2 m^{-2} h^{-1}$ ($CE_{N_2O}$ and $CE_{CH_4}$, respectively), using a GWP of 28 for $CH_4$ and 265 for $N_2O$ over 100 years (*IPCC, 2013*).

$$GWP_{\text{soil export}} = CE_{CO_2} + CE_{CH_4} \times 28 + CE_{N_2O} \times 265 \qquad (4)$$

Where $GWP_{\text{soilexport}}$ is the Global Warming Potential of soil export (kg ($CO_2$) $hm^{-2}$); CE is accumulative emission flux of GHG.

All datasets were processed with Origin 20.0 software for figure construction. One-Way ANOVA was implemented in the software of SPASS 22.0 to compare the significant differences in GHG fluxes and soil environmental parameters from different elevations and land-use sites. To ensure that residuals from different statistical analyses were normally distributed and homogeneous, variables were log transformed. As an alternative, Kruskal-Wallis H test was used to deal with those with non-normal distributions ($CH_4$ fluxes and $N_2O$ fluxes in different land uses). A value of $P < 0.05$ was used to infer statistical significance.

The type of ordination method appropriate for GHG emissions across a gradient length was determined through detrended correspondence analysis (DCA) (*Borcard, Legendre & Drapeau, 1992*). Because gradient length was smaller than 4 in this study, redundancy analysis (RDA) was chosen to analyze the relationships between GHG metrics and environmental parameters. All explanatory variables and GHG metrics were standardized with the function: $(y_i - y_{min})/(y_{max} - y_{min})$. RDA was performed using the R package "vegan", and validated by Monte Carlo methods with 999 permutations (*Shan et al., 2019*).

## RESULTS

### GHG fluxes from different elevations during the growing season

During the discharging period, $CH_4$ fluxes from dry lands at four elevations ranged from $-0.48$ to $2.44$ mg m$^{-2}$ h$^{-1}$ (Fig. 4A). $CH_4$ fluxes from paddy fields at three elevations in 2015 ranged from $-0.87$ to $10.13$ mg m$^{-2}$ h$^{-1}$ between April to September (Fig. 4B).

No significant differences in $CH_4$ and $CO_2$ emissions from different elevations were found (Table 2). However, $CO_2$ emissions in dry lands and paddy fields varied in different elevations during growing season (Figs. 4C and 4D). Mean $CO_2$ fluxes from dry lands

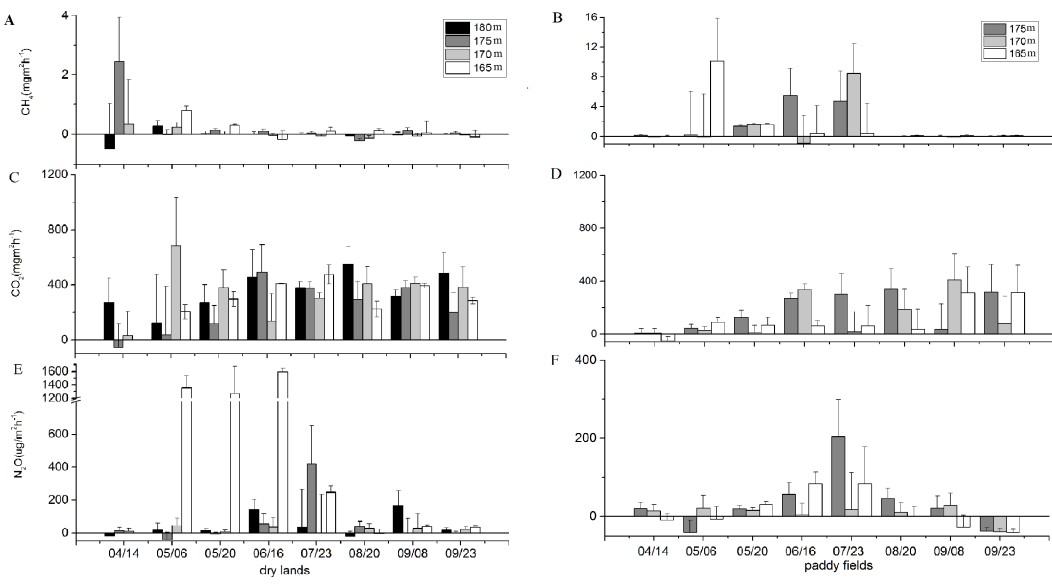

**Figure 4** CH$_4$, CO$_2$ and N$_2$O emissions of different elevations in Wuyang Bay in the growing season in 2015. (A) CH$_4$, (C) CO$_2$ and (E) N$_2$O emissions in dry lands; (B) CH$_4$,(D) CO$_2$ and (F) N$_2$O emissions in paddy fields.

were estimated as $358.75 \pm 139.78$ mg m$^{-2}$ h$^{-1}$ (mean $\pm$ standard error) (180 m), $231.95 \pm 189.22$ mg m$^{-2}$ h$^{-1}$ (175 m), $342.91 \pm 196.22$ mg m$^{-2}$ h$^{-1}$ (170 m) and $328.31 \pm 100.73$ mg m$^{-2}$ h$^{-1}$ (165 m). Meanwhile, mean CO$_2$ fluxes from paddy fields were measured as $161.36 \pm 272.79$ mg m$^{-2}$ h$^{-1}$ (175 m), $183.68 \pm 293.36$ mg m$^{-2}$ h$^{-1}$ (170 m) and $311.85 \pm 539.82$ mg m$^{-2}$ h$^{-1}$ (165 m). A decreasing trend was showed along with the rising elevation in paddy fields.

N$_2$O fluxes varied from $-39.92$ to $1,594.14$ µg m$^{-2}$ h$^{-1}$, and the highest N$_2$O flux was from dry lands at the elevation of 165 m in June (Fig. 4E). The mean N$_2$O fluxes from dry lands at an elevation of 165 m ($627.04 \pm 721.16$ µg m$^{-2}$ h$^{-1}$) were significantly higher ($F = 4.809$, $P = 0.008$) than those at an elevation of 180 m ($44.73 \pm 71.51$ µg m$^{-2}$ h$^{-1}$), 175 m ($59.82 \pm 148.26$ µg m$^{-2}$ h$^{-1}$) and 170 m ($22.67 \pm 15.56$ µg m$^{-2}$ h$^{-1}$) (Table 2). Mean N$_2$O fluxes at different elevations in paddy fields ranged from 2.88 to 20.03 µg m$^{-2}$ h$^{-1}$, but the means were associated with very high variability (Fig. 4F). Therefore, there might be visible trends in the overall means, but they exhibited a very weak or no statistical difference in N$_2$O flux at different sampling elevations. Concretely, the mean of N$_2$O flux at an elevation of 175 m ($20.03 \pm 22.41$ µg m$^{-2}$ h$^{-1}$) in paddy fields tended to be higher than those values at 170 m ($9.52 \pm 25.77$ µg m$^{-2}$ h$^{-1}$) and 165 m ($2.88 \pm 16.45$ µg m$^{-2}$ h$^{-1}$).

## GHG flux from different land use types

From April to September in 2016, the mean CH$_4$ fluxes from paddy fields ($0.35 \pm 0.72$ mg m$^{-2}$ h$^{-1}$) tended to be higher than those from the dry lands ($0.036 \pm 0.09$ mg m$^{-2}$ h$^{-1}$) and grasslands ($0.08 \pm 0.20$ mg m$^{-2}$ h$^{-1}$) (Table 3, Fig. 5A). However, there was no

Wang et al. (2020), *PeerJ*, DOI 10.7717/peerj.8503

**Table 2** CH$_4$, CO$_2$ and N$_2$O emissions and soil environmental parameters of different elevations in Wuyang Bay in 2015.

| | Elevations (m) | Soil volumetric water content (VWC) (%) | Soil gravimetric water content (GWC) (%) | Soil temperature (°C) | NO$_3^-$-N (mg kg$^{-1}$) | NH$_4^+$-N (mg kg$^{-1}$) | Total carbon (TC) (g kg$^{-1}$) | Total Nitrogen (TN) (g kg$^{-1}$) | Dissolved organic carbon (DOC) (gkg$^{-1}$) | CH$_4$ (m$^{-2}$ h$^{-1}$) | CO$_2$ (m$^{-2}$ h$^{-1}$) | N$_2$O (μg m$^{-2}$ h$^{-1}$) |
|---|---|---|---|---|---|---|---|---|---|---|---|---|
| | 180 | 52.50 ± 9.15[a] | 25.33 ± 4.77[a] | 23.82 ± 4.86[a] | 16.17 ± 18.80[a] | 27.34 ± 18.80[a] | 7.71 ± 0.68[a] | 0.93 ± 0.21[a] | n.a. | −0.036 ± 0.21[a] | 358.75 ± 139.78[a] | 44.73 ± 71.51[a] |
| | 175 | 63.42 ± 5.64[b] | 37.56 ± 7.23[b] | 24.84 ± 4.32[a] | 18.71 ± 9.81[a] | 18.91 ± 9.81[a] | 17.52 ± 6.39[b] | 1.59 ± 0.34[b] | n.a. | 0.34 ± 0.86[a] | 231.95 ± 189.22[a] | 59.82 ± 148.26[a] |
| Dry lands | 170 | 51.40 ± 9.53[a] | 27.19 ± 4.55[a] | 24.5 ± 4.15[a] | 15.79 ± 10.44[a] | 16.05 ± 10.44[a] | 14.34 ± 5.79[b] | 1.31 ± 0.42[b] | n.a. | 0.032 ± 0.17[a] | 342.91 ± 196.22[a] | 22.68 ± 15.56[a] |
| | 165 | 42.56 ± 7.78[c] | 28.64 ± 4.7[c] | 26.79 ± 4.39[a] | 57.56 ± 26.98[b] | 38.23 ± 26.98[a] | 12.90 ± 2.73[b] | 1.41 ± 0.23[b] | n.a. | 0.16 ± 0.32[a] | 328.31 ± 100.73[a] | 627.04 ± 721.16[b] |
| | 175 | 67.39 ± 19.41[ab] | 64.11 ± 20.25[a] | 25.62 ± 2.55[a] | 3.29 ± 5.56[a] | 29.36 ± 29.54[a] | 19.81 ± 3.03[a] | 1.88 ± 0.28[a] | 0.28 ± 0.076[a] | 0.53 ± 1.49[a] | 161.36 ± 272.79[a] | 20.03 ± 25.77[a] |
| Paddy fields | 170 | 67.02 ± 20.27[a] | 56.69 ± 17.78[b] | 26.36 ± 2.95[ab] | 2.91 ± 6.90[a] | 13.92 ± 13.64[b] | 15.91 ± 2.48[b] | 1.38 ± 0.28[b] | 0.22 ± 0.046[b] | 0.19 ± 0.55[a] | 183.68 ± 293.36[a] | 9.52 ± 35.63[a] |
| | 165 | 66.26 ± 20.58[b] | 59.89 ± 20.14[ab] | 27.80 ± 2.54[b] | 0.43 ± 0.51[b] | 20.44 ± 19.33[ab] | 19.88 ± 2.34[a] | 1.73 ± 0.15c | 0.22 ± 0.043[b] | 0.96 ± 1.89[a] | 311.85 ± 539.82[a] | 2.88 ± 16.45[a] |

**Notes.**

Values are means ± standard error.

n.a., not analyzed

a,b,c different letters in the same column means statistical significance ($P < 0.05$).

Wang et al. (2020), *PeerJ*, DOI 10.7717/peerj.8503

**Table 3** CH$_4$, CO$_2$ and N$_2$O emissions and soil environmental parameters of different land uses at the elevation of 170 m in Wuyang Bay in 2016.

| Land uses | Soil volumetric water content (VWC) (%) | Soil gravimetric water content (GWC) (%) | Soil temperature (°C) | NO$_3^-$-N (mg kg$^{-1}$) | NH$_4^+$-N (mg kg$^{-1}$) | Total carbon(TC) (g kg$^{-1}$) | Total Nitrogen(TN) (g kg$^{-1}$) | Dissolved organic carbon (DOC)(g kg$^{-1}$) | CH$_4$ (mg m$^{-2}$ h$^{-1}$) | CO$_2$ (mg m$^{-2}$ h$^{-1}$) | N$_2$O (μg m$^{-2}$ h$^{-1}$) |
|---|---|---|---|---|---|---|---|---|---|---|---|
| Dry lands | 46.64 ± 14.04$^a$ | 30.38 ± 9.82$^a$ | 26.08 ± 2.68$^a$ | 6.02 ± 4.06$^a$ | 4.86 ± 3.48$^a$ | 13.52 ± 2.24$^a$ | 1.30 ± 0.20$^a$ | 0.22 ± 0.06$^a$ | 0.036 ± 0.09$^a$ | 266.91 ± 180.79$^a$ | 50.61 ± 40.71$^a$ |
| Paddy fields | 60.50 ± 23.67$^b$ | 49.89 ± 20.42$^b$ | 26.49 ± 2.63$^a$ | 1.74 ± 2.44$^b$ | 16.38 ± 18.53$^b$ | 16.01 ± 2.70$^b$ | 1.44 ± 0.24$^b$ | 0.23 ± 0.05$^a$ | 0.35 ± 0.72$^b$ | 151.63 ± 213.55$^a$ | 22.36 ± 34.89$^b$ |
| Grass fields | 49.07 ±.71$^a$ | 39.87 ± 64$^c$ | 26.29 ± 3.0$^a$ | 3.43 ± 3.23$^b$ | 6.13 ± 4.44$^a$ | 16.90 ± 3.74$^b$ | 1.52 ± 0.29$^b$ | 0.26 ± 0.057$^b$ | 0.08 ± 0.20$^a$ | 395.43 ± 283.32$^b$ | 50.42 ± 75.18$^{ab}$ |

**Notes.**

Values are means ± standard error.

a,b,c different letters in the same column means statistical significance ($P < 0.05$).

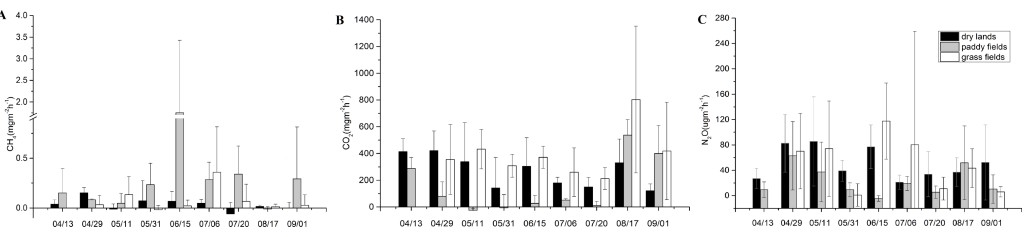

**Figure 5** CH$_4$, CO$_2$ and N$_2$O emissions from drylands, paddy fields and grass fields at the elevation of 170 min in the growing season in 2016. (A) CH$_4$, (B) CO$_2$, and (C) N$_2$O emissions

significant difference in CH$_4$ fluxes of these three land uses. Monthly CH$_4$ fluxes varied greatly among the sampling period, and the CH$_4$ flux exhibited lowest level in August 17th. In paddy fields, the maximum CH$_4$ flux was observed in July, and it was the source of methane throughout the sampling time.

CO$_2$ fluxes were positive in dry lands and grasslands, but negative in paddy fields in the early growth stage (Fig. 5B). In the summer, CO$_2$ fluxes in paddy fields increased to positive levels as the temperature rose. On August 17th, CO$_2$ emissions in the three land use types increased. The mean CO$_2$ fluxes from grasslands ($395.43 \pm 283.32$ mg m$^{-2}$ h$^{-1}$) were significantly higher ($F = 9.958$, $P < 0.001$) than those values from the dry lands ($266.91 \pm 180.79$ mg m$^{-2}$ h$^{-1}$) and paddy fields ($151.63 \pm 213.55$ mg m$^{-2}$ h$^{-1}$) (Table 3).

N$_2$O fluxes were positive in these three land use types from April to September in 2016 (Fig. 5C). With respect to N$_2$O emission in different land use types, there was significant difference ($H = 19.556$, $P = 0.005$) between paddy fields and dry lands. Due to high variability in measured values, no significant difference was found between grass fields and paddy fields in N$_2$O flux. However, the overall means might exhibit a distinct trend. For instance, the mean N$_2$O fluxes from paddy fields ($22.36 \pm 34.89$ μg m$^{-2}$ h$^{-1}$) was approximately half of those values from dry lands ($50.61 \pm 40.71$ μg m$^{-2}$ h$^{-1}$) and grasslands ($50.42 \pm 75.18$ μg m$^{-2}$ h$^{-1}$) (Table 3).

## Diurnal emissions of GHG emissions during the growing season

Table 4 exhibited the diurnal emissions of GHGs during different seasons in cropland of the riparian zone. It was shown that TDF of CO$_2$ in paddy field in September was much higher than that in other seasons, while TDF of CH$_4$ and N$_2$O flux in paddy field in September were much lower than that in other seasons. According to the lines of diurnal variations of GHG emissions in Fig. 6 and GHGs flux in Table 4, GHG flux from 9:00 to 11:00 basically can represented the mean emission of the day.

Diurnal changes in CO$_2$ and N$_2$O fluxes from dry lands and paddy fields were observed during the growing season (Fig. 6). CO$_2$ andN$_2$O fluxes were low at night and high during the day in dry lands, and the same trajectory was observed in daily air temperature. The daily emission peaks of CO$_2$ and N$_2$O fluxes were all at 2–4 pm, and the amplitude (the difference between the highest and lowest values within a day) of diurnal variations of the CO$_2$ andN$_2$O fluxes were various in different season. Diurnal variations in CH$_4$ emissions were not obvious both in dry lands and paddy fields in different seasons. Moreover, no

**Table 4** Diurnal emissions and total daily flux (TDF) of GHGs ($CH_4$, $CO_2$ and $N_2O$) during different seasons in cropland of Wuyang Bay.

| | | Sep 2015 | May 2016 | July 2016 | Sep 2016 |
|---|---|---|---|---|---|
| Dry lands | Mean $CH_4$/mg m$^{-2}$ h$^{-1}$ | $-0.015 \pm 0.11$ | $-0.018 \pm 0.071$ | $0.025 \pm 0.092$ | $0.00026 \pm 0.016$ |
| | $CH_4$ flux from 10 am/mg m$^{-2}$ h$^{-1}$ | $-0.018$ | 0.044 | 0.047 | 0.014 |
| | Mean $CO_2$/mg m$^{-2}$ h$^{-1}$ | $404.11 \pm 114.68$ | $475.977 \pm 209.88$ | $146.18 \pm 45.51$ | $291.98 \pm 255.34$ |
| | $CO_2$ flux from 10 am/mg m$^{-2}$ h$^{-1}$ | 316.52 | 672.90 | 163.82 | 163.95 |
| | Mean $NO_2$/µg m$^{-2}$ h$^{-1}$ | $52.99 \pm 18.54$ | $68.44 \pm 21.76$ | $38.34 \pm 21.31$ | $120.71 \pm 29.73$ |
| | $NO_2$ flux from 10 am/mg m$^{-2}$ h$^{-1}$ | 19.97 | 67.34 | 62.89 | 97.23 |
| Paddy field | Mean $CH_4$/mg m$^{-2}$ h$^{-1}$ | $0.00079 \pm 0.11$ | $0.38 \pm 0.68$ | $0.061 \pm 0.12$ | $-0.0030 \pm 0.025$ |
| | $CH_4$ flux from 10 am/mg m$^{-2}$ h$^{-1}$ | $-0.020$ | 0.032 | 0.0074 | 0.010 |
| | Mean $CO_2$/mg m$^{-2}$ h$^{-1}$ | $237.78 \pm 340.43$ | $2.35 \pm 107.18$ | $50.12 \pm 46.35$ | $284.66 \pm 316.18$ |
| | $CO_2$ flux from 10 am/mg m$^{-2}$ h$^{-1}$ | 147.44 | $-75.36$ | 32.26 | 179.13 |
| | Mean $NO_2$/µg m$^{-2}$ h$^{-1}$ | $-3.67 \pm 16.72$ | $10.44 \pm 16.83$ | $11.25 \pm 20.89$ | $2.24 \pm 2.05$ |
| | $NO_2$ flux from10 am/mg m$^{-2}$ h$^{-1}$ | $-9.64$ | 13.84 | 4.73 | 0.20 |
| Dry lands | TDF $CH_4$/mg m$^{-2}$ d$^{-1}$ | $-0.37$ | $-0.44$ | 0.61 | 0.0063 |
| | TDF $CO_2$/mg m$^{-2}$ d$^{-1}$ | 9,698.64 | 11,423.34 | 350,831 | 700,741 |
| | TDF $NO_2$/µg m$^{-2}$ d$^{-1}$ | 1,271.77 | 1,642.59 | 920.11 | 289,710 |
| Paddy field | TDF $CH_4$/mg m$^{-2}$ d$^{-1}$ | 0.019 | 9.19 | 1.45 | $-0.07$ |
| | TDF $CO_2$/mg m$^{-2}$ d$^{-1}$ | 5,706.81 | 56.49 | 1,202.96 | 683,193 |
| | TDF $NO_2$/µg m$^{-2}$ d$^{-1}$ | $-88.19$ | 250.64 | 269.97 | 53.88 |

**Notes.**
Values are means ± standard error.

obvious fluctuating regularity of the $N_2O$ diurnal emission could be observed during the observation period in paddy fields.

Mean $CO_2$ diurnal emission values in dry lands amounted to $404.11 \pm 114.68$ mg m$^{-2}$ h$^{-1}$ in September in 2015, $475.92.11 \pm 209.88$ mg m$^{-2}$ h$^{-1}$ in May, $146.18 \pm 45.51$ mg m$^{-2}$ h$^{-1}$ in July and $291.98 \pm 255.33$ mg m$^{-2}$ h$^{-1}$ in September in 2016(Table 4). Mean $CO_2$ diurnal emission values in paddy fields amounted to $237.78 \pm 340.43$ mg m$^{-2}$ h$^{-1}$ in September in 2015, $2.35 \pm 107.78$ mg m$^{-2}$ h$^{-1}$ in May, $50.12 \pm 46.35$ mg m$^{-2}$ h$^{-1}$ in July and $284.66 \pm 316.18$ mg m$^{-2}$ h$^{-1}$ in September in 2016. In August 2016, $CO_2$ flux increased, the amplitude of diurnal variation was also quite high, and the peak emission were above 1,000 mg m$^{-2}$ h$^{-1}$ in dry lands and paddy fields.

Mean $N_2O$ diurnal emission values in dry lands amounted to $120.71 \pm 29.73$ µg m$^{-2}$ h$^{-1}$ in September 2016, which was higher than in other sampling occasions. However, there were no differences in the amplitude of diurnal variation in different seasons. Mean $N_2O$ diurnal emission values in paddy fields amounted to $-3.67 \pm 16.72$ in September 2015 and $2.24 \pm 2.05$ µg m$^{-2}$ h$^{-1}$ in September 2016 which was lower than that at other sampling times, and the amplitude of diurnal variation in September was also much lower than other seasons.

### Cumulative emission of GHG in different land use types

Mean $CH_4$ fluxes of the riparian zone from 2015 to 2016 in this study was $0.47 \pm 1.53$ mg m$^{-2}$ h$^{-1}$(Figs. 4 and 5). Cumulative emission fluxes of $CH_4$ in paddy fields were much higher than those in dry lands and grass fields (Table 5). Whereas, cumulative emission

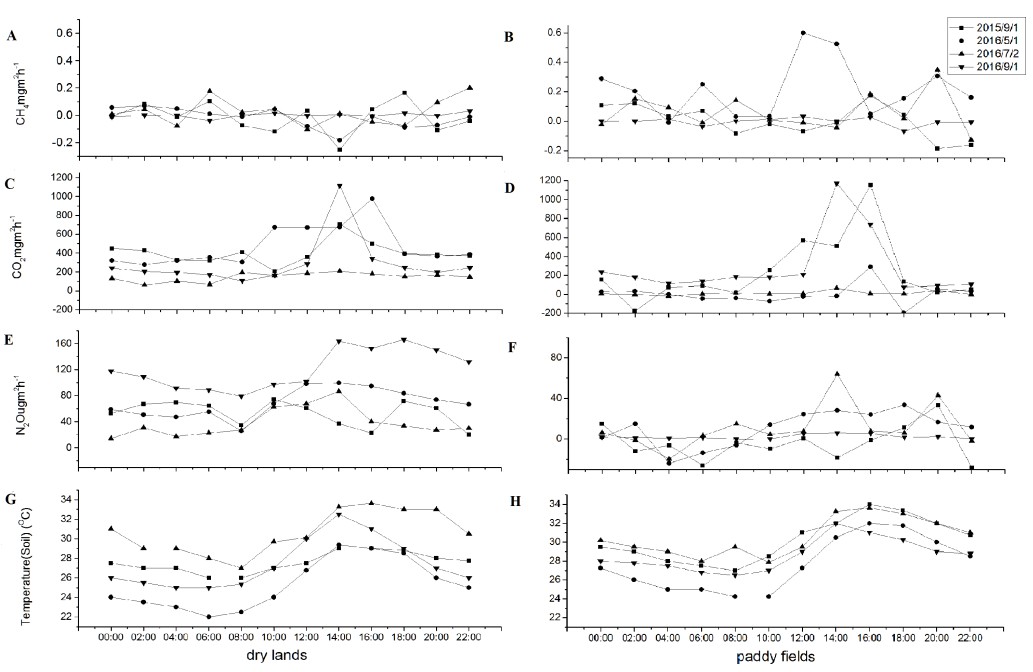

**Figure 6** Diurnal emissions of $CH_4$, $CO_2$ and $N_2O$ in different seasons during the growing period. (A) $CH_4$, (C) $CO_2$, (E) $N_2O$ and G) Temperature(soil) in dry lands; (B) $CH_4$, (D) $CO_2$, (F) $N_2O$ and H) Temperature (soil) in paddy fields

**Table 5** Accumulative emission fluxes of $CH_2$, $CO_2$ and $N_2O$ and the global warming potential of soil surface in Wuyang Bay over the growing season.

| Land uses | $CH_4$ fluxes (kg hm$^{-2}$) | | $CO_2$ fluxes (kg hm$^{-2}$) | | $N_2O$ fluxes (kg hm$^{-2}$) | | $CH_4$ fluxes equivalent (kg($CO_2$) hm$^{-2}$) | | $N_2O$ fluxes equivalent (kg($CO_2$) hm$^{-2}$) | | GWP (kg($CO_2$) hm$^{-2}$) | |
|---|---|---|---|---|---|---|---|---|---|---|---|---|
| | 2015 | 2016 | 2015 | 2016 | 2015 | 2016 | 2015 | 2016 | 2015 | 2016 | 2015 | 2016 |
| Paddy fields | 75.69 | 13.39 | 561,974 | 10,724.03 | 1.16 | 1.11 | 211,920 | 374.84 | 306.52 | 293.18 | 804,546 | 11,392.06 |
| Dry lands | 4.72 | 1.59 | 13,694.61 | 11,726.90 | 2.26 | 2.15 | 132.15 | 44.42 | 599.31 | 569.26 | 14,426.07 | 12,340.58 |
| Grass fields | na | 2.95 | na | 17,700.26 | na | 2.04 | na | 82.47 | na | 539.94 | na | 18,322.66 |

fluxes of $CO_2$ and $N_2O$ in dry lands and grass fields were higher than those in paddy fields. GWP was 8,045.46 kg ha$^{-1}$ in 2015 and 113,392.06 kg ha$^{-1}$ in 2016 in paddy fields, while 14,426.07 kg ha$^{-1}$ in 2015 and 12340.58 kg ha$^{-1}$ in 2016 in dry lands during the growing season. GWP in grasslands (18,322.66 kg hm$^{-1}$) in 2016 was higher than that in cropland, which is beneficial to the high flux of $CO_2$ (17,700.26 kg hm$^{-1}$). $CO_2$ emission from the sampling site contributed the most to GWP, $CH_4$ emission contributed the second most, and $N_2O$ emission contributed the least. In the whole riparian zone of the study area, $CO_2$ emissions represented the greatest proportion of GWP (>70.0%).

## Influencing factors of GHG emissions

Air temperatures through sampling time ranged from 22.0 °C to 40.0 °C. Soil temperature at 165m of paddy fields was significantly higher ($F = 6.113$, $P = 0.003$) than other elevations.

The mean content of $NO_3^--N$ in soil at 165m (57.66 mg kg$^{-1}$) in dry lands was significantly ($F = 4.809$, $P = 0.008$) higher than that of other elevations. The mean TC ($F = 5.585$, $P = 0.004$) and TN ($F = 50.293$, $P = 0.00$) at 180 m (unflooded area) were significantly lower than others, while the TC and TN at 170 m for paddy fields were lower than others. It is clearly shown that significant variations in soil VWC exhibited an increasing trend with the increase of elevations in riparian zone (Table 2).

Mean soil temperatures in riparian zone for dry lands, paddy fields and grass fields were 24.93 °C, 26.44 °C and 22.29 °C through sampling time, respectively. Soil MWC in dry lands (30.38%) was significantly ($F = 9.892$, $P < 0.001$) lower than that in paddy fields (49.89%) and grass fields (39.87%), while the content of $NO_3^--N$ in dry lands (6.02 mg kg-1) was higher and $NH_4^+-N$ (4.86 mg kg-1) was lower than that in the other two land uses. TC, TN and DOC in dry lands were lower than those in paddy fields and grasslands (Table 3).

The redundancy analysis (Fig. 7) explored several potential impact factors that may regulate three types of GHG emissions based on the data obtained from all sites in the study area. Strengths of the correlations between environmental factors and the first two axes were summarized in Table 6. The first two axes accounted for 37.30% of GHGs variation in all data (2015–2016), 45.6% in dataset from paddy fields, 53.96% in dataset from dry lands and 26.18% in dataset from grass fields. The RDA ordinations showed that the flux of $CO_2$ positively correlated with ST, but negatively correlated with soil GWC in all data (Fig. 7A). Soil temperature and soil GWC were positively ($r = 0.994$) and negatively ($r = -0.998$) correlated with the first axis, respectively. The flux of $N_2O$ positively related to AT ($r = 0.544$) and soil $NO_3^--N$ ($r = 0.906$), which were positively correlated with the second axis. Soil $NO_3^--N$ ($r = 0.999$) and AT ($r = 0.962$) were positively correlated with the first axis, and soil GWC was positively ($r = 0.952$) correlated with the second axis in dry lands (Fig. 7B). With respect to the paddy fields, soil temperature ($r = 0.999$) and air temperature ($r = 1.0$) were positively correlated with the first axis. While soil GWC ($r = 0.999$) and DOC ($r = 0.999$) negatively correlated with the first axis (Fig. 7C). Increase of $CO_2$ fluxes was stimulated by increase of soil T and air T, but with the decrease of soil GWC and DOC. In grass fields, $CO_2$ fluxes was negatively correlated with soil VWC which was positively correlated with the first axis ($r = -0.896$) and second axis ($r = -0.444$) (Fig. 7D).

## DISCUSSION

### The contribution of cultivation to GHG emissions

This study shows that soil of the bay riparian zone acts as a weak $CH_4$ and $N_2O$ source, but a strong $CO_2$ source during the growing season.

Mean $CH_4$ fluxes of the riparian zone from 2015 to 2016 in this study (0.47 ± 1.53 mg m$^{-2}$ h$^{-1}$) were similar to values reported for other wetland catchments (0.0∼2.22 mg m$^{-2}$ h$^{-1}$) (*Badiou et al., 2011*; *Finocchiaro, Tangen & Gleason, 2014*; *Yang et al., 2014*). The production of $CH_4$ in various eco-systems is closely related to the anaerobic environment, which is beneficial to the methanogen metabolic activities (*Zhou et al., 2014*; *Zhang et al.,*

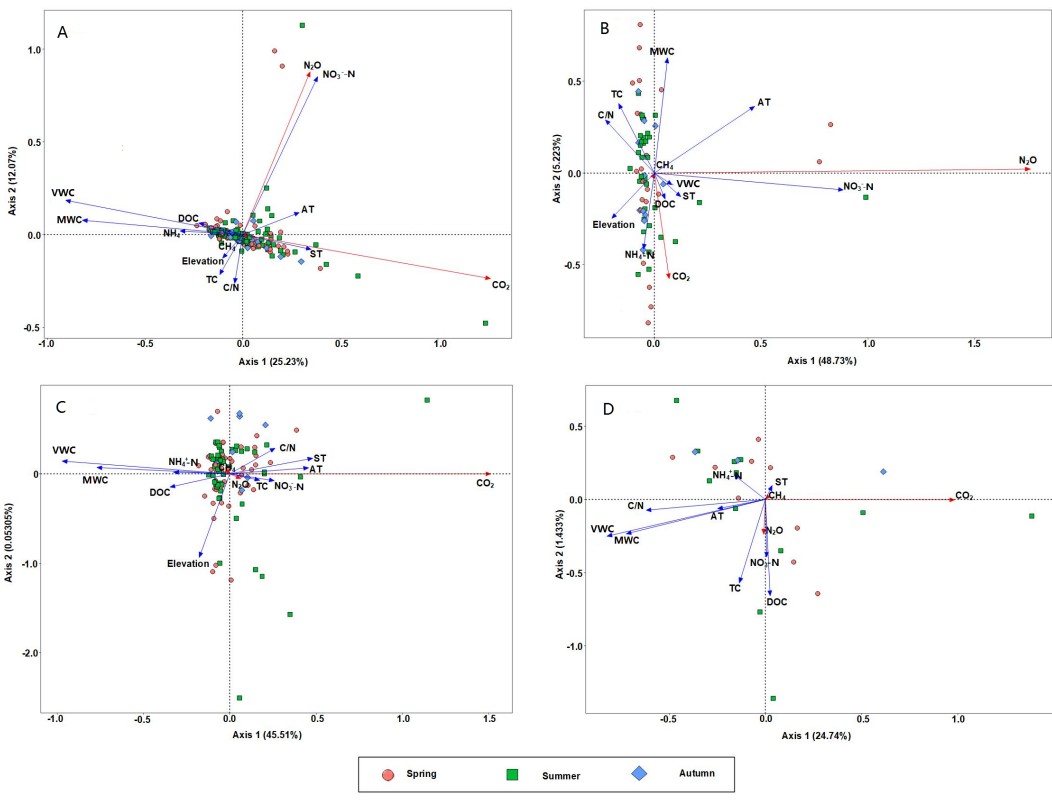

**Figure 7  Redundancy analysis between CH₄, CO₂ and N₂O emissions and environmental parameters.** (A) All data (2015–2016); (B) paddy fields; (C) uplands; (D): grasslands (2016). Red arrows are GHG emissions, and blue arrows are environmental parameters. In this figure, TC, soil total carbon; TN, soil total nitrogen; $NO_3^-$, nitrate nitrogen; $NH_4^+$, ammonia nitrogen; ST, soil temperature; VWC, soil volumetric water content; MWC, soil mass water content, AT, air temperature; DOC, dissolved organic carbon.

*2018*). Soil water content, which affects the methane consuming capacity of methane-oxidizing bacteria by changing the transmission rate and the soil environmental condition (*Koh, Ochs & Yu, 2009*), probably was a reason of the significant difference in the CH₄ fluxes among the paddy fields, dry lands and grasslands.

In recent years, the riparian zone is believed be an important source of N₂O emissions, because it might be the key area of nitrogen transformation in the whole biogeochemical system (*Groffman, Gold & Addy, 2000*; *Wang et al., 2006*). Incomplete denitrification under anaerobic conditions was considered to be the major reason for the production of N₂O (Wilcock and Sorrell, 2008; *Zhou et al., 2014*). Since N₂O eventually produces N₂ in anaerobic conditions, N₂O emissions from paddy filed were much lower than those from dry lands and grasslands. N₂O emission is limited when soil humidity completely saturates to values >60% in paddy fields (*Davidson et al., 2000*; *Finocchiaro, Tangen & Gleason, 2014*) and nitrifiers convert ammonium into nitrates to promote the N₂O flux in an aerobic soil environment after the wet soil has dried (*Kaiser & Heinemeyer, 1996*; *Yang et al., 2017*). Therefore, N₂O fluxes in paddy fields showed a double-peak pattern before planting and after crop harvest in 2016 (Fig. 5). The N₂O emission in grasslands without fertilizer might

Table 6 Correlation matrix showing the relationship between RDA axes and significant ($P < 0.05$) environmental variables, and the results of the RDA analysis.

| Axis variable | All data | | Dry lands | | Paddy fields | | Grass fields | |
|---|---|---|---|---|---|---|---|---|
| | Axis 1 | Axis2 | Axis 1 | Axis2 | Axis 1 | Axis2 | Axis 1 | Axis2 |
| Elevation | | | | | −0.831 | −0.557 | | |
| Soil VWC | −0.997 | 0.080 | | | −0.999 | 0.036 | −0.896 | −0.444 |
| Soil WC | −0.998 | −0.061 | 0.304 | 0.952 | −0.999 | 0.029 | | |
| AT | 0.839 | 0.544 | 0.962 | 0.275 | 1.000 | −0.001 | | |
| ST | 0.994 | −0.110 | | | 0.999 | 0.025 | | |
| $NO_3^-$ | 0.424 | 0.906 | 0.999 | −0.020 | | | | |
| TC | −0.469 | −0.883 | | | | | | |
| DOC | | | | | −0.999 | −0.031 | | |
| Eigenvalues | 0.003 | 0.002 | 0.018 | 0.002 | 0.006 | 0.000 | 0.010 | 0.001 |
| Cumulative percentage variance of species-environment relation (%) | 25.23 | 37.30 | 48.73 | 53.96 | 45.51 | 45.56 | 24.74 | 26.18 |

Notes.
In this figure,
TC, soil total carbon; $NO_3^-$, nitrate nitrogen; ST, soil temperature; VWC, soil volumetric water content; MWC, soil mass water content; AT, air temperature; DOC, dissolved organic carbon.

be due to surface runoff or sufficient substrate from high mineralization rates under the dry-wet alternating actions in the riparian zone (*Muller et al., 2004*).

Cumulative fluxes of GHGs (Table 5) showed that $CO_2$ emission dominated the GHG fluxes when the $CH_4$ and $N_2O$ fluxes were calculated as $CO_2$ equivalents during the growing season, which is comparable to the results archived in *Audet et al. (2013)* and *Zhou et al. (2017)*. Litter from soil surface, root residues and exudates in splits are the main source of soil organic matter in terrestrial ecosystems (*Gleixner, 2005*; *Gonzalez-Meler et al., 2018*). The high organic matter in riparian soil along with water level fluctuation created slow decomposition of external organic matter and microorganisms (*Schimel et al., 2011*), and insoluble organic carbon transferred to inorganic carbon through mineralization resulting in the increase of $CO_2$ emissions (*Andrews & Schlesinger, 2001*). Soil TC and DOC in croplands were lower than in grass fields (Table 3), probably due to the intensive interference of anthropogenic activities in cropland which may loosen the soil, increase the soil permeability, and hence lead to the easy run-off of soil DOC (*Wang et al., 2005*). These may also be the reasons why grass fields that grow herbaceous plants without tillage have the greatest emissions of $CO_2$, and have the highest levels of GHG fluxes.

**The influence of water level fluctuation and tillage on GHG emission**
Although no significant variations in $CH_4$ emission were found at different elevations, the mean $CH_4$ fluxes at 180 m ($-0.036\,\mathrm{mg\,m^{-2}\,h^{-1}}$) (unflooded area) were lower than at other elevations (flooded area), and soil TC and TN were also lower than that at the riparian zone (Table 2), which were similar to results from *Tangen, Finocchiaro & Gleason (2015)*. However, an increasing trend did not appear as the elevation decreased in the riparian zone, which was different from the research on grasslands in TGR by *Chai et al. (2017)*.

The variations in the $CH_4$ emission in different land uses correlated with the changes in the soil water content (Fig. 7), but the $CH_4$ emission in dry lands at different elevations did not show the same decreasing trend with soil water content in riparian zone (Table 2). That means different inundation duration varied the soil water content in dry lands, but did not lead to the variations of GHG emissions, and the variations in different seasons may correlate with various characteristics, such as soil temperature and human interference specially, as time proceeds.

$N_2O$ fluxes in the research area showed great spatial and temporal variability. The highest $N_2O$ flux from dry lands was from the elevation of 165 m (May to June), which reached 1,594.14 $\mu g\ m^{-2}\ h^{-1}$ in June. $N_2O$ fluxes in dry lands showed a one-peak pattern after a flooding period. The flux of $N_2O$ positively related to soil $NO_3^--N$ (Fig. 7). In paddy fields and dry lands, $N_2O$ fluxes were positively related to air temperature and negatively related to soil TC. Application of farm manure provided the substrate for soil microbial growth, and promotes nitrification and denitrification reaction (*Bouwman, 1996*; *Zheng et al., 2004*). The major reason for the highest observed flux of $N_2O$ might be tillage and fertilization in dry lands at 165 m for vegetable planting in May.

Fluxes of $CO_2$ in this study showed an insignificant increasing trend along the elevations, which was different with the study in Miyun reservoir (*Yang, 2016*) and Fuling TGR (*Xu et al., 2017*). Crop harvest exposes soil at the surface. In addition, water–table position and temperature, which influences the development of aerobic and microbial activities, would influence the potential of GHGs emmission (*Glatzel, Basiliko & Moore, 2004*; *Waddington, Rotenberg & Warren, 2001*). The research site went through a draught period from 10th to the 24th of August, 2016. After crop harvest, soil VWC, GWC, ST and $NH_4^+-N$ decreased, and $NO_3^--N$ increased on August 17th, 2016 (Figs. S1 and S2). Therefore, crop harvest combined with continual dry and hot weather in August increased the $CO_2$ emission especially in paddy fields.

## Differences of GHG fluxes between the bay riparian zone and other zones

GHG fluxes measured in this typical tributary bay riparian zone of the TGR were within the ranges of previous research across different sites (Table 7). Because of the higher soil moisture content, $CH_4$ flux in paddy fields was higher than that in grass fields and dry lands in the riparian zone, but lower than that in paddy fields in Hubei and SiChuan (*Lin et al., 2000*; *Han et al., 2005*). Based on the data in Table 7 (*Shiau et al., 2016*; *Yu et al., 2010*), $N_2O$ fluxes in grass fields in riparian zone in this research were higher than in other natural wetlands due to high soil $NO_3^--N$ and soil redox conditions, which favored mineralization in periodic water level fluctuation (*Badiou et al., 2011*; *Shiau et al., 2016*). In addition, the application of nitrogen fertilizer would increase the denitrification rate and denitrification product ratio ($N_2O/N_2$) (*Xing, 1998*). So $N_2O$ fluxes in cropland system were higher than that in wetland without tillaging. Discrepancies in $N_2O$ and $CO_2$ fluxes in different croplands are likely related to the different amount of fertilizer, as well as geographic location, weather, and timing and intensity of sampling (*Stelzer et al., 2011*).

**Table 7  $CH_4$, $CO_2$ and $N_2O$ fluxes from different lands.**

| Location | Land type | $CH_4$ fluxes ($mg\ m^{-2}\ h^{-1}$) | $CO_2$ fluxes ($mg\ m^{-2}\ h^{-1}$) | $N_2O$ fluxes ($\mu g\ m^{-2}\ h^{-1}$) | Ref. |
|---|---|---|---|---|---|
| Yetang Creek,TGR,China | corn fields | −0.13∼0.77 | 85.09∼420.78 | 2.39∼151.11 | |
| | paddy fields | −0.0098∼4.57 | −24.56∼537.47 | −30.71∼101.76 | Current study |
| | fallow grasslands | −0.017∼0.36 | 212.41∼803.74 | 0.94∼117.61 | |
| Beijia Creek, TGR,China, | dike-ponds | 0.26–7.69 | 171.24–275.72 | −30.0∼100 | *Zhou et al. (2017)* |
| Lijiaba, TGR,China | fallow grasslands | −0.36∼0.086 | 124.96∼865.92 | 1.76∼34.32 | *Li et al. (2016)* |
| | corn fields | −0.13∼0.052 | 321.64∼599.72 | 0.11∼701.36 | |
| Miyun reservoir | riparian zone | 0.05∼6.4 | −98∼2274 | −136.6–381.8 | *Yang (2016)* |
| Carteret county, North Carolina | marsh Wetland | −0.22 ∼0.31 | −166.62∼323.3 | −70.0∼75.36 | *Shiau et al. (2016)* |
| Sanjiang Plain,Northern China | marsh wetland | −0.16 ∼1.71 | – | −1.24∼4.65 | *Yu et al. (2010)* |
| Outer coastal plain, US | restored agricultural wetland | −0.04∼0.32 | 91.38∼740.24 | −6.28∼792.85 | *Morse, Ardón & Bernhardt (2012)* |
| Praire Pothole ,Canada | wetland | 0.21∼2.22 | – | 2.20∼3.27 | *Badiou et al. (2011)* |
| Glaciated plains,US | cropland in wetland | 1.08∼111.25 | – | 30.42∼43.75 | *Tangen, Finocchiaro & Gleason (2015)* |
| | cropland upland | 0.0042∼0.20 | – | 20.46∼33.75 | |
| Croping system, US | corn fields | – | 91.75∼917.5 | −39.25∼425.20 | *Adviento-Borbe et al. (2010)* |
| Croping system, US | corn fields | 0.0072∼0.0117 | – | 8.12∼121.13 | *Amos, Arkebauer & Doran (2005)* |
| Freeman farm, US | corn-soybean fileds | −0.37∼0.88 | 1,16724∼2806.7 | −170.23∼448.74 | *Panday & Nkongolo (2015)* |
| SiChuan, croping system, China | corn fields | −0.11∼3.95 | 485.83∼703.63 | 19.64 | *Su (2016)* |
| | paddy fields | 0.00∼17.8 | 232.47∼378.77 | | |
| HuBei, cropping system | paddy fields | 4.39∼15.6 | | | *Lin et al. (2000)* |
| SiChuan, cropping system | paddy fields | 1.66∼17.51 | | | *Han et al. (2005)* |

GHG fluxes from this study were in line with values reported for other typical tributary bay riparian zones in TGR. $CH_4$ emission fluxes in the natural wetland (e.g., marsh wetland) and man-made wetland (e.g., paddy fields) were higher than that from the different types of dry lands. GHG emissions in reservoirs are mainly released at the initial periods after the impoundment due to the abrupt release of nutrient substances in the flooded lands, the elevation of microbial activities, and the decomposition of unstable carbon matters, such as soils, litters, and leaves (*Tremblay et al., 2005*). As it is in the early development of TGR riparian zone, there may be in a decreasing trend in GHG emissions in future.

## CONCLUSIONS

Riparian zones in tributary bays of the TGR are generally a source of GHGs. $CO_2$ emission in this kind of zone mainly contributed to the GHG emissions, partly due to the high organic carbon input from the reservoir catchment, and they should be paid more attention in the future research.

Different farming practices likely changed the humidity, carbon and nitrogen of soil. GHG emissions in TGR were affected by soil temperature, soil water content and soil fertility. Riparian zone reclamation lowered $CO_2$ emission, and increased $N_2O$ emission. Higher soil VWC led to a higher $CH_4$ emission in paddy fields than in dry lands and grasslands. Anthropogenic activities caused a decrease of soil TC, and the decrease of GHG emission from croplands in the riparian zone. Farming was a main influence (e.g., tillage and fertilization), and caused higher $N_2O$ fluxes at 165 m especially. The high $N_2O$ fluxes produced from tillage and fertilizer suggest that, in order to potentially mitigate GHG emissions from the riparian zone, farming practices in dry lands at low water levels (below 165 m) should be paid more attention.

Because of the resident farming activities, TGR cropland seemed to have no spatial effect along the elevation gradient. No significant difference of the amplitude of diurnal variations of GHGs in different seasons was found. However, there was a short-term increasing $CO_2$ and decreasing $N_2O$ emission, and the diurnal variation amplitude of $N_2O$ was also much lower after draining in paddy fields. These spatial–temporal characteristics in paddy fields together with the seasonal water-level fluctuation shall be taken into account in future studies to further explore the underlying mechanisms controlling GHG emission kinetics.

## ACKNOWLEDGEMENTS

We thank the farmers near the field site for assistance of sampling. Special thanks are given also due to Yi Jiang and Zhimei Liu for their help in the field and laboratory.

### Funding

This work was supported by the Natural Science Foundation of China (No. 41701247; No. 51609229; No. 41771266), the Fundamental and Frontier Research Project of Chongqing (No. cstc2017jcyjAX0415), and the Chongqing Science and Technology Commission (No.

cstc2017jcyjAX0241). The funders had no role in study design, data collection and analysis, decision to publish, or preparation of the manuscript.

### Grant Disclosures

The following grant information was disclosed by the authors:
Natural Science Foundation of China: 41701247, 51609229, 41771266.
Fundamental and Frontier Research Project of Chongqing: cstc2017jcyjAX0415.
Chongqing Science and Technology Commission: cstc2017jcyjAX0241.

### Competing Interests

The authors declare there are no competing interests.

### Author Contributions

- XiaoXiao Wang conceived and designed the experiments, performed the experiments, analyzed the data, prepared figures and/or tables, authored or reviewed drafts of the paper, and approved the final draft.
- Ping Huang conceived and designed the experiments, prepared figures and/or tables, and approved the final draft.
- Maohua Ma and Shengjun Wu conceived and designed the experiments, authored or reviewed drafts of the paper, and approved the final draft.
- Kun Shan analyzed the data, prepared figures and/or tables, and approved the final draft.
- Zhaofei Wen performed the experiments, prepared figures and/or tables, and approved the final draft.

### Data Availability

The raw data are available in the Supplementary File.

### Supplemental Information

Supplemental information for this article can be found online at http://dx.doi.org/10.7717/peerj.8503#supplemental-information.

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
