# Peer review of "Greenhouse gas emissions from riparian zone cropland in a tributary bay of the Three Gorges Reservoir, China"

_PeerJ, doi:10.7717/peerj.8503_

## Round 0.1 · original submission · Major Revisions

All components of your manuscript must be improved to a large extent in order to be considered for publication. The reviewers are offering you detailed recommendations. Once you have made all corrections and improvements as required, you are welcome to re-submit the manuscript.

Reviewer 1 ·

Basic reporting

• The methods are not sufficient and provide almost no details required to evaluate the results of this paper. The methods must be greatly modified before a proper review of this paper can be performed. Specific concerns and examples are provided below, although this list is not exhaustive.
o Statistical software programs are specified, but citations are required.
o Diurnal data are presented and discussed, but there is no mention of such sampling in the methods. In fact, the methods state that GHG sampling was conduced between 9 and 11 am. Moreover, the methods state that this time period was representative of mean daily fluxes, but provide no evidence to back up this statement. Also, calculation of diurnal flux is not presented, and the units are in m-2 hr-1, which do not seem like diurnal emissions which typically would be mass/area/day?
o Soils data are presented and discussed, but there also is no mention of such sampling (how,when,where).
o The location, number, and distribution of static chambers is not provided. How many were samples per site, how were they distributed, etc.
o The methods state environmental parameters, and presumably fluxes, were analyzed and that P<0.05 was ‘considered acceptable.’ There is no mention of what statistical tests were used, or how they were structured. This is unacceptable for a peer-reviewed article. Also, it is not clear what ‘considered acceptable’ means when referred to the p-value. This should read something similar to P values <0.05 were used to infer statistical significance, etc.
o Similarly, for the RDA, no information is provided as how the RDA was structured or which options were selected in Canoco. Further, it appears that the soils data were used as covariates for the GHG flux data, but it is not clear if this is appropriate due to the lack of methodologies. For example, fluxes were measured ~monthly over 2 years. Were soils collected at similar times and locations, or were they a one-time sample. If so, the soil properties could differ, especially among years, and the may not be appropriate for relating to flux measurements collected at different times. Moreover, were they collected from the same locations, what depths were they collected from, how were they analyzed, etc.?
o Since opaque chambers were used, CO2 flux could represent both soil and plant respiration if vegetation was present within the chambers. Please clarify in the methods whether there was vegetation present in the chambers, and if so possibly adjust your discussion and conclusions accordingly.
o A formula is provided that describes the calculation of cumulative flux, but it is not clear what the cumulative flux represents. For example, were the values from the sample dates simply summed, or were the data extrapolated for the number of days between sample events; if so, how was this done?
o One of the soil variables is ‘humidity.’ It is not clear what this refers to (humidity typically is water vapor in the atmosphere?), and there are no units provided on Table 3. Since it appears that water contents also is presented in Table 3, I assume that this simply isn’t a terminology issue where water content is referred to as humidity. Either way, these variables need to be defined in the methods, and sampling methods, etc. need to be presented.
o The methods sate that ‘an area from 165-175 m was chosen’ and that sites were classified into different elevations “165-180”; these two do not match.
• As detailed below, I’m not sure how informative comparisons are of GHG flux between wetland and upland sites, as the abiotic conditions that regulate flux are quite difference among these environments. Moreover, it is not clear which sites were inundated and dry during sampling, and this information would be quite helpful for analyses and interpretation of results.
• Figures 1, 4, & 6 are not referenced in the text.
• Number the figures and tables in the order that they are presented. For example, the first reference for Table 2 follows the reference for Table 4.
• All figure and table legends/captions should be improved to include relevant information (what/where/when) and to make them stand-alone. For example, define abbreviations such as TN from table 1 and include units for all variables. Also, provide relevant information as to when and where data were collected, etc. Information pertaining to statistical differences also should be included and described. For example, it is not clear which year is represented by figures 4 and 5, nor is it clear which ‘treatments’ were significantly different.
• Table 1: it is not clear where these data are from or when and how they were collected; there is no methodology presented.
• Why in some cases (e.g., ghg flux from land use types) are only data from 2016 presented?
• In the results, when discussion trends are relative fluxes between treatments make sure to present the associated data/numbers and reference related figures/tables.
• In the Discussion, there are several instances where you make statements/speculations on things that you did not test; this should be avoided. If making a general statement that is supported by other literature, make sure to make it clear that you are not referring to results of your study, but general theories/processes supported by other research. Following are a few examples:
o Soil water content affected the methane consuming capacity of methane-oxidizing bacteria by changing the transmission rate and the soil environmental condition (Koh et al., 2009), resulting in a significant difference in the CH4 fluxes (P<0.05) among the paddy fields uplands and grasslands.
o Crop harvest combined with continual dry and hot weather in August increased the CO2 emission especially in paddy fields
o Application of farm manure provided the substrate for soil microbial growth, promoting nitrification and denitrification reaction and leading to an increase in N2O emissions
o Soil CO2 fluxes peaked in August, indicating that both soil and root respiration increased with the increase in air temperature
o N2O fluxes in grasslands in this paper were higher than in other wetlands owing to the surface flow from neighboring farms. In addition, N2O fluxes in wetlands were lower than that in croplands because of fertilization.
• The raw data could be formatted to facilitate examination or importing into statistical software. Also, metadata are required, as well as information pertaining to calculations, etc.
• Language:
o The paper is fairly well written, but the English could be improved. I suggest an additional edit with the goal of improving the language and work selection. Following are a couple of examples:
 It is not clear what ‘unsystmatically cultivated’ refers to
 It is not clear what ‘speculative reclamation in the riparian zones’ refers to
 …frequent anthropogenic activities against periodic water-level fluctuation; against doesn’t seem to be appropriate here
 The 3 aims were to compare flux differences; I would say they were to compare fluxes, not differences
o Once you define an acronym, use it thereafter. For example, after you define GHG, use GHG rather than writing out greenhouse gas.
o Be sure to check standard formatting such as ‘et al’ should be ‘et al.’, etc. Also, double check for missing/extra spaces, etc.
o Check all of your unit presentations, for example, kg·ha-2 should be kg·ha-1 (line 139).
o Consistent use of terminology related to the study sites and land uses would be beneficial to avoid confusion. For example, it appears that croplands and uplands are used interchangeably, which causes confusion. Further, grasslands typically are considered uplands, which once again, adds some confusion. Lastly, riparian zone also is used, but it is not always clear which samples are included. I suggest using clearly-defined terminology for each land use type and then being consistent with the usage.

Experimental design

• It is well accepted in the GHG discipline that methane fluxes differ greatly between wetlands and uplands; thus, I’m not sure the comparisons between the two areas are relevant or provide any valuable information.
• The Methods were not to sufficient to appropriately assess the experimental design.
• Inferences are made in this paper that land-use practices, etc. effected N2O flux, despite the fact that these hypotheses weren’t tested. It has been well established that N2O flux is to a large degree driven by soil moisture, but this wasn’t even examined here. It would seem that in a riparian area characterized by fluctuating water levels, that soil moisture would be a primary variable of interest that likely has great influence on GHG flux, especially N2O. Thus, I suggest exploring these relationships, especially since Table 3 suggests that WC differed by elevation.

Validity of the findings

• The conclusions state that differences in N2O flux were due to agriculture and fertilization. I would argue that soil moisture or water levels may play and equal or greater role and should be examined more thoroughly.
• The paper references submergence and submerged time, but there are no data or analyses.
• The authors state that crop harvest increased CO2 flux, but this statement isn’t strongly supported by data and analyses; I suggest providing greater support for such statements.
• Data and analyses should be provided for conclusions such as “Farming practices changed humidity, C, N of soil” etc.
• Since comparing wet and dry sites, differences could be due to hydrology rather than land use. I suggest examining all abiotic covariates among the land-uses and elevations, in addition to fluxes.

Additional comments

• GHG fluxes from the transitory, riparian zone of reservoirs are lacking; thus, the goals of the paper address a relevant topic.
• In its current form this paper is not acceptable for publication. After considerable revision, however, the work represented here could result in an acceptable manuscript. To do this, the methods require significant modification, and the focus of the study should be refined. Also be sure to make sure all study goals and conclusions are supported by data, and that alternative explanations for results are acknowledged. Due to the deficiencies with the methods, as well as concerns with the results and conclusions, I point out major aspects to be revised, but do not provide comprehensive comments throughout the paper.

Reviewer 2 ·

Basic reporting

The article objectives are well explained.
The site description is detailed.

Experimental design

It is necessary to detail the chromatograph method, as collum, oven temperature, detectors used…
Detail the figures of the merit of the chromatograph method.
Detail the methods used to measure SH, ST, WC, TC, TN, nitrogen compounds…

Validity of the findings

Line 152. Try to use new IPCC reference and updated GWP
Line 159: detail CANOCO
What is the difference between the RDA and PCA (principal components analysis)?
The results are well presented and also discussed.

Additional comments

The article can be published after minor corrections.

---

## Round 0.2 · Major Revisions

A careful evaluation of the revised manuscript was carried out. Improvements have been made based on the previous comments/suggestions addressed by the reviewers. However, some important improvements and corrections are still required. The reviewer provided suggestions and guidance for major revisions that I believe will greatly improve the paper, if the authors take the time to address them sufficiently. After that, tha authors are welcome to submit the manuscript once more.

Reviewer 1 ·

Basic reporting

• The English language is adequate, but the grammar, sentence structure, and tense could be improved. Special attention should be payed to singular versus plural tense. A few examples are: LN 114: rice straw was cut/corn straw was left…; LN 133: September represents…; LN 328: did not show…
• The Introductory and background information and the references provide adequate background and are appropriate.
• The structure of the paper follows general standards and is acceptable.
• Table and figure legends must be improved to make them stand-alone and to aid in interpretation by a reader. Include all relevant information as to ‘what/where/when.’ For example, rather than say …collected from study sites in this experiment in 2015, specify where the experiment was conducted, etc. Also, make sure to define all acronyms, etc. (e.g., GHG, TGR). The overall goal should be to include sufficient information so a reader would not have to search the text to identify ACRONYMS, treatments, etc.
• Figure 1: The caption says water level of riparian zone in Pengxi River while the text (e.g., LN 68) says water level in TGR. I assume the figure is reservoir level, but please clarify.
• Figure 2: There are multiple panels, with two labeled as A and C; there is no B. Further, the panels are not presented or described in the caption. Each panel, including the image, should be described and referenced (here and elsewhere). Further, there are 2 legends in the figure, and the formatting is not consistent between them. Lastly, the reference to Fig. 2 on LN 102 should be moved to LN 98 where the study area is referenced; the figure does not provide information pertaining to dates of agricultural activities discussed in LN 102.
• Figure 4: Each panel should be described/referenced in the caption. Also, information in the legend should be specified, including units, (i.e., elevation). For panel A and F, appropriate breaks/scaling should be used so the data can be clearly observed. For example, the bars in panel A are not visible or readily interpretable.
• Numbers indicating statistical significance (e.g., Table 2 and elsewhere) should be superscript (here and elsewhere)
• Figure 5: The terminology in the caption (upland/grasslands) does not match the legend (dry lands/grass fields). Also, the panels are not described in the caption. Panel B does not have a label for the Y axis.
• Table 4. The caption states ‘diurnal variation…’ but the Table also includes TDF; modify the caption for completeness and accuracy. Define TDF, replace upland with dry lands. May and July are not abbreviated so do not require periods
• Table 5: variables (e.g., VWM, MWC, ST) need to be defined and units provided for all (e.g., VWC). Also, flux units need more specific to differentiate between CH4/CO2/N2O fluxes and CO2-equivalents (GWP).
• Table 6: similar issues to other tables, plus formatting of references is inconsistent.
• Figure 7: shows WC while the text uses MWC. There are other instances of this throughout the paper as well. Change WC to MWC throughout.
• Figure 8: I suggest making the X and Y axes scales even for each panel. The X and Y axes titles are insufficient (define RDA1, RDA2, etc.) and the % within are not explained. What dates were considered spring, summer, and autumn? Are the points individual samples, means by date, etc.??
• The raw data (supplement) is incomplete and presented in a manner that is difficult to decipher. The supplement only includes GHG data from 2016? The supplement should provide all data collected and analyzed for the study by site, land use, elevation, and date.
• Some raw data is supplied, but there are no metadata or variable descriptions to help decipher the data. For example, definitions and units should be provided for all variables. Also, with some work a user could figure out the elevation using the column headings, but land-use type isn’t obviously included. In the end, these data could be structured better and more information must be provided for a user to replicate the results. As an example, I suggest including columns that identify date, site, elevation, land use, and concentration of GHG’s for the sample replicates. It also would be useful to present the final flux value calculated from these raw data, as well as the model used to calculate flux for each sample. Or, describe these models and the process in the methods of the paper.
• While some of the raw GHG data is supplied, the soils and covariate data are not; thus, results of the analyses cannot be replicated.

Experimental design

• The experimental design seems acceptable, although additional details are required to confirm this (see general comments).

Validity of the findings

• GHG data, especially CH4, almost always are highly skewed and require a data transformation to meet model assumptions of normality. If the data were transformed prior to running the ANOVA this should be detailed. If not, it may be appropriate to state that data were distributed normally and didn’t require transformation or something like that.
• The methods (LN 129-131) describe gas samples collected every 10 minutes for 30 minutes. The methods (typically regression) used to determine change in concentration over 30 minutes are not provided. Please describe the methods/models used.
• LN 144: how was the concentration of GHG in standard atmospheric conditions measured/determined?
• LN 147: it is not clear what dc/dt is; please define each term. As presented, it also isn’t clear how change in concentration was determined. The methods here seem like there was a time 0 and time 30 sample, but samples were collected at 10, 20, and 30 minutes? Please clarify methods.
• LN: 153-154; define ‘t’
• LN 165: should WC be MWC (here and elsewhere)?
• LNs 177-183: How was the ANOVA parametrized? Were repeated samples over time treated as repeated measures or unique samples? How was CANOCO parametrized for the RDA (i.e., which options were specified in CANOCO)? Were any of the data transformed? Are means presented in the tables arithmetic means or LS means from the ANOVA?
• Providing an ANOVA results table for all analyses would be helpful (F and P values, DF, etc.)
• It is a bit unclear, but the methods description makes it appear as if there is no replication among the land uses? Essentially it is written as if there was a single chamber in dry lands and paddy lands for each elevation, and the seasonal means are based off of these. Based on the error bars in Figure 4 I assume that there were multiple sample paddys and dry lands sampled at each elevation? Please clarify this (e.g., describe number of sample sites), and present the sample size for all analyses and field sampling. This raw data also should be supplied in the supplement. Presentation of sample size should be included in the tables/figures/analyses.
• LN’s 186-188: the ranges for CH4 presented here do not match the data provided in Figure 4, which shows means exceeding 2 for dry lands and 10 for paddy fields. Also, the text references April–August while the figure provides data through September.
• Ln 199: the N2O flux value does not match the value in Table 2; please re-check all values in text and tables for accuracy.
• LNs 197-198: I would caution against making such statements since there were no significant differences and the means are associated with very high variability. In essence, there may be visible trends in the overall means but they are very weak owing to the high variation. Comment applies to similar instances elsewhere (e.g., LNs 215-218).
• LN218: The value (se) presented for dry lands does not match the value from Table 3; please re-check all values in text and tables for accuracy.
• LNs 220-223: The author’s highlight ‘obvious’ diurnal changes and relate them to ‘similar’ patters in temperature. While there is some variation, the trends are much weaker than indicated. There are some instances where CO2 and N2O show higher daytime values, but there are also many instances where they do not. Also, the relations between trends in GHG and temperature are not ‘similar trajectories,’ rather I would say that some of the higher flux values coincide with peak temperatures but drop off much more rapidly than temperature. In the end, I suggest modifying conclusions so as to not overstate the trends in your data. Further, you could perform formal correlations between temperature and flux if you would like to demonstrate/evaluate these relations.
• LNs 232-234: the authors state that CO2 flux increased and the amplitude of diurnal variation also was high and peak emissions were above 1000… I would caution against attributing the high value to harvest since there are similar peaks from other ‘non-harvest’ time periods. Are there other factors that could contribute to these observations?
• LNs 252-255: I suggest including the GWP for each gas in Table 5, along with the cumulative fluxes. This will allow a reader to determine the relative importance of each gas. As it stands only the total GWP is presented along with normal fluxes for each GHG.
• LN 257: I don’t see the value of this single-sentence paragraph? Moreover, it isn’t correct. First, decreased/increased between which date and August, July? And if so, why single out this time period. Second, ST didn’t decrease in August. Third, there are variable trends in the means (up and down) over time. If the goal here is simply to present these data I wouldn’t discuss trends unless you are going to analyze them formally or explore them further. These data also may be more appropriate as a table, appendix, or supplementary file.
• LNs 258-269: Why only mean soil temp for 2 of the 3 land uses? I don’t see the point in a statistical comparison for air temperature since it was a localized study area where temperature should be consistent? Results are provided by land use in some instances, elevation in others, and yet by elevation only for single land uses for others; why the inconsistency and what is the purpose of these comparisons? Also, trends are discussed but the data aren’t presented in a table or figure? This entire section is confusing and needs to be modified and supported with tables/figures if it is even retained.
• LNs 271-274: why only this detailed information for 8a and not the other panels?
• LN 282: what does P value represent and why 0.1 (methods state 0.05 as cutoff)? Were all axes/relations significant?
• LNs 270-282: for the RDA a table showing the correlations between axes and variables would be helpful.

Additional comments

This revised manuscript is improved from the original submission, and the authors attempted to address most of the previous review comments. I believe that the story the authors are trying to tell can be supported by this study, but major revisions to the manuscript still are required to meet standards of a peer-reviewed journal. Specifically, the methods require greater detail, presentation of results must be improved, conclusions must be supported by the data, and the formatting must be improved. The data supplement also required additional information and revision. While not exhaustive, this review provides comments, suggestions, and guidance to address weakness of the paper that, if properly addressed, should result in a paper suitable for publication.

• It may be an artifact of converting to .pdf for review but there are numerous instances throughout the paper where spaces between words is required (e.g.,’ Wuyang Bayin’ from Figure 4 caption and differntseasons from Table 4 caption). If this is not an artifact of conversion please check the paper carefully.
• LN 63: the acronym NCCC is not needed as it isn’t used elsewhere.
• TGR, the acronym for Three Gorges Reservoir, should be defined after the first use in LN 68, rather than LN 69 (2nd use).
• LN 70-71: this sentence is not clear. 632 km2 were submersed is clear, but it is not clear what the ‘14.9% of total affected lands’ refers to?
• Include sample size (n) for data in Table 1
• Ln 92: You are not comparing fluxes of different water-level elevations, you are simply comparing different elevations, correct? The riparian zone was not submerged during the study, correct?
• LN 111-112, what is ‘a-1’ in the fertilizer amount, year-1??
• Values presented in the text (e.g., LN 205) should match values in the table (Table 3) and not be rounded differently.
• In LNs 211-212 you say that CO2 increased for all land uses after harvest; this implies a relationship between harvest and flux. Why would grassland CO2 increase since there would presumably be no harvest? Also, what was harvested in August, rice, corn, or both, and was the harvest before or after the mid-August sample event? Could this trend be associated with something other than harvest (temperature/precipitation, soil moisture)?
• LN 228: reference Table 4 after presenting these values.
• LN 236 and 238: Specify September of 2016…
• LNs 238-239: September of 2015 was lower than this value.
• LNs 243-244: the values from 9-11 may be similar to the daily mean, but it is difficult to tell simply by looking at these graphs because of the variability. If this determination is important, simply calculate hourly and daily means in a table to show whether this is the case or not. Further, this statement doesn’t logically fit here; I suggest moving it elsewhere, such as the first paragraph of this section discussion diurnal variation.
• LNs 285-286: single-sentence paragraphs aren’t appropriate. Suggest combining this with the following paragraph.
• LN 287: where is this mean value from? New data that hasn’t been previously presented in the results should not be introduced in the Discussion. It is not even clear how this mean was calculated or what time period it represents, etc.
• LN 288: what were the values (i.e., ranges) reported for other catchments?
• LNs 293-294: the P-value cutoff does not need to be presented in the Discussion, and the Table should be referenced for these flux values. Or, simply reference the table rather than the numbers since they have already been presented.
• Line 301: replace soil humidity with term used previously throughout the paper.
• LNs 304-305: where is this double peak shown or demonstrated? Figure/Table?
• LN 308: should the table referenced be Table 5 rather than Table 2? Also, the GWP for individual gases is not provided to support this statement. Only the overall GWP is provided.
• LN 316: TC was not sig. lower in croplands than grasslands (see Table 3)
• LNs 319-320: this sentence is confusing. It seems as though you are saying grasslands have similar attributes as croplands. I think you mean that the reasons stated before is why cropland have lower emissions? Grasslands would have higher emissions because they did not have those characteristics.
• LNs 324-325: what is meant by submerged time and where was this trend examined???
• LN 334: where are these values from? It is not appropriate to present new data in the Discussion. Also, reference tables/figures in this section when discussion data/trends.
• LN 346: Figure 7 does not show drought or any climate data. It does show some soil moisture data, but this drought prior to August isn’t evident and August 24 is not shown?
• LNs 355-359: why compare grasslands to natural wetlands? Wouldn’t differences in N2O be expected? Also, is fertilizer applied to grasslands (358-359)?
• LN 360-362: what cropland and wetland systems are being compared here?
• LN 364: what natural wetland/marsh is being referred to here?
• LN 382: you say that to potentially mitigate GHG…farming practices should be considered. But, this study showed that in terms of GWP, N2O was least important. So is this really important?

---

## Round 0.3 · Minor Revisions

The manuscript still requires many small improvements before it can be considered for publication in PeerJ.

Reviewer 1 ·

Basic reporting

a. I commend the authors for publishing in a second language, and while the writing is improved from previous versions, the language is not quite up to standards for an English language peer reviewed journal. I have provided comments for improvement, and suggest an additional edit by English language expert to improve readability and comprehension.
b. Overall the background and introduction provide sufficient context for the study and the literature is relevant. I do provide comments, however, where more appropriate citations could be provided in certain instances.
c. The overall structure of the manuscript conforms to standards.
d. The figures and tables have improved compared to previous versions, but still require modification in some instances (see specific comments).
e. The raw data are supplied and appear complete.

Experimental design

a. The research is within Scope of the journal and can help address the knowledge gap associated with GHG fluxes from riparian areas of large reservoirs. The experimental design and methods appear sound and the description of the methods and procedures is sufficient.

Validity of the findings

a. Overall the findings and conclusions are reasonable. However, there are a few instances where the language should be modified where statements are not supported by analyses and/or data (see specific comments). Moreover, some of the results and trends should be re-evaluated based on comments provided.

Additional comments

a. Comments and suggestions are provided here as well as in an annotated version of the .pdf. When making recommendations on language and phrasing, I only provide the comment for the first instance or two with the expectation that the authors will address similar issues throughout the paper. Specifically, most of the editorial suggestions are provided in the introduction section and should be applied throughout the manuscript where appropriate.
b. Overall the authors made an effort to address all comments from the previous reviews.
c. There are a few instances where the authors attribute results to a factor or variable without formally testing a hypotheses or statistically examining relations; most of these, however were rectified since the previous version of the manuscript. In these instances, I suggest simply modifying the language so as to not make statements that are not supported by data. Following is an example: Line 28: the authors state that CO2 was the main contributor because of high OC. However, this hypothesis was not tested and there could be other confounding factors. Simply saying ‘…most likely because of…’ would be more appropriate.
d. LNs 62-65: This sentence is a bit confusing, it says that agricultural emissions = 11% of 820 (~90), but that emissions from rice and agricultural uses were 374. What is the difference between agricultural emissions and rice and agricultural land uses?
e. LN 123: here and elsewhere I would reword this. GHG emission were not measured, rather gas samples were collected and GHG emissions were calculated.
f. LN 163: typically when referring to the molecule NO3-/NH4+ are used; when referring to nitrate/ammonia nitrogen the following are commonly used: NO3—N/NH4+–N. I suggest considering these.
g. Lines 180-186: Here is says that the data were log transformed and analysed using a 1-way ANOVA. It also says that a Kruskal-Wallis H test was used to deal with non-normal distributions. If multiple tests were used, you need to identify which data were not normally distributed and which test was associated with each analyses presented in the results. As currently described it is not clear which test was used with which data?
h. LN 197-199: Why are only the data from 2015 shown here? The methods state that sampled were collected monthly during 2015-2016?
i. LN 204: at first use in the text and in all table/figure captions define what the error estimates are (e.g., +/- se, sd)?
j. LN 211: here and elsewhere, please present the degrees of freedom along with the F and P values.
k. LN 219: here and elsewhere, I suggest changing the language from “GHG flux differences…” to simply “GHG fluxes from…” Overall, you are not presenting differences, you are presenting fluxes and testing for differences, of which there are none in many instances.
l. LN 222: Table 3 only is referenced for the CH4 data; I suggest referencing it for the other gases as well.
m. Line 230: the P value is missing the 3rd digit value. Here and elsewhere make sure all F and P values have all of the required digits/information.
n. LN 234: The F value is not presented with the P value.
o. LNs 235-238 and Table 3. The statistical results for N2O among the 3 land uses appears to be wrong. The dry lands and grass fields have nearly identical values, while the paddy field value is much less. However, sig. differences are presented for the dry lands and grass fields but not the dry lands and paddy fields? Based on the high variability and mean values I do not see how this is possible if the sample sizes are somewhat equivalent and the test was run properly. Please check the results and interpretation of this test, and if necessary, provide the sample size information.
p. LNs 246-248: I do not see the trends identified here. In some cases the daytime values may be higher, but in others they are not. Also, unless you run a formal experiment or test you cannot say that the trajectory was influenced by air temperature; at best you could say that they appear correlated. I suggest rewording this discussion and if you are going to identify trends make sure they are demonstrated and be specific. For example, if 2 dates show a slight trend and 2 do not I’m not sure you can identify the trend. If you do, I would then identify the dates where you observe any trends and the dates where you do not. This would allow for you to address the trends or lack of trends in the Discussion.
q. Ln 249: Throughout the diurnal section ‘amplitude’ is used, but it is not always clear what it refers to. Are you using amplitude to refer to the difference between the highest and lowest values within a day? Or, does it mean something else? I suggest providing a brief definition of amplitude after the first use to avoid confusion.
r. Ln 268: What does this mean value represent? Which samples were includes and how was it calculated? Is this presented in a table anywhere? Overall, what is the implication or relevance?
s. LN 271: when you say higher than in paddy field, is this qualitative or quantitative? Similar comments elsewhere (e.g., LN 275)
t. LN 271-273: I am not clear what this sentence means or refers to. Please clarify or provide further details, refer to values in a table, etc.?
u. LN 276: it is unclear what “beneficial to the high flux of CO2” means?
v. LNs 278-279: This is a bit unclear, do you mean that CO2 emissions represented the greatest proportion of GWP of the 3 gases?
w. LNs 286-288: the phrase ‘significant variations in soil water content exhibited an increasing trend’ if fairly confusing. Moreover, as best as I can tell the data in table 2 do not show such a trend. Please modify this sentence and identify the data that show this trend?
x. LNs 295-310: describing the relations between environmental variables and each axis isn’t very informative. Rather, I suggest focusing on the important relations between the significant environmental variables and the GHG fluxes.
y. Ln 314: you say that the study area is a strong CO2 source. It may be worthwhile to mention that you are only looking at CO2 emissions and not ecosystem uptake (e.g., NPP). Basically, the net flux may be different?
z. LN 316-317. This comparison to values from the literature is okay. However, the systems referenced here are small, ephemeral mineral soil wetlands. thus, it may be beneficial to also include estimates from more comparable systems (e.g., large reservoirs)??
aa. LNs 319-322: you did not test the transmission rate…etc. and thus can not say that if resulted in a change. I suggest modifying this and similar language to reflect whether you are speculating or discussing probably relationships or whether you are discussion something that you tested scientifically.
bb. LN 395: what natural wetland are you referring to? This study did not sample natural wetlands, only croplands, grasslands, and paddy’s??
cc. LNs 405-405: CH4 also required organic carbon, so I’m not sure this is the primary reason the CO2 emissions were higher. Is it not more likely due to environmental and biotic conditions that influence flux? Or at least some combination?
dd. All figure and table captions, identify what the data and errors are: e.g., mean GHG fluxes +/- se, etc.
ee. Figure 4: I suggest using the same format for the x axes of the left and right panels (1 vs. 2-digit months). Make consistent among all figures/tables.
ff. Table 5: The caption could be modified to better describe the data presented.

Annotated reviews are not available for download in order to protect the identity of reviewers who chose to remain anonymous.

---

## Round 0.4 · accepted · Accept

The place where the figures shall be inserted in the text has not been shown in your manuscript. Therefore, PeerJ will decide the best place where they shall be placed.